# Cold sensitivity of TRPA1 is unveiled by the prolyl hydroxylation blockade-induced sensitization to ROS

Takahito Miyake[1], Saki Nakamura[1], Meng Zhao[1], Kanako So[1], Keisuke Inoue[2], Tomohiro Numata[2,3], Nobuaki Takahashi[2], Hisashi Shirakawa[1], Yasuo Mori[2], Takayuki Nakagawa[1,4] & Shuji Kaneko[1]

Mammalian transient receptor potential ankyrin 1 (TRPA1) is a polymodal nociceptor that plays an important role in pain generation, but its role as a cold nociceptor is still controversial. Here, we propose that TRPA1 can sense noxious cold via transduction of reactive oxygen species (ROS) signalling. We show that inhibiting hydroxylation of a proline residue within the N-terminal ankyrin repeat of human TRPA1 by mutation or using a prolyl hydroxylase (PHD) inhibitor potentiates the cold sensitivity of TRPA1 in the presence of hydrogen peroxide. Inhibiting PHD in mice triggers mouse TRPA1 sensitization sufficiently to sense cold-evoked ROS, which causes cold hypersensitivity. Furthermore, this phenomenon underlies the acute cold hypersensitivity induced by the chemotherapeutic agent oxaliplatin or its metabolite oxalate. Thus, our findings provide evidence that blocking prolyl hydroxylation reveals TRPA1 sensitization to ROS, which enables TRPA1 to convert ROS signalling into cold sensitivity.

[1] Department of Molecular Pharmacology, Graduate School of Pharmaceutical Sciences, Kyoto University, 46-29 Yoshida-Shimoadachi-cho, Sakyo-ku 606-8501, Japan. [2] Department of Synthetic Chemistry and Biological Chemistry, Graduate School of Enginnering, Kyoto University, Katsura Campus, Nishikyo-ku 615-8510, Japan. [3] Department of Physiology, Graduate School of Medical Sciences, Fukuoka University, Nanakuma 7-45-1, Jonan-ku 814-0180, Japan. [4] Department of Clinical Pharmacology and Therapeutics, Kyoto University Hospital, 54 Shogoin-Kawahara-cho, Sakyo-ku 606-8507, Japan. Correspondence and requests for materials should be addressed to T.Na. (email: tknakaga@kuhp.kyoto-u.ac.jp).

Animals possess diverse sensors for various latent dangers in the environment. The sensors are mainly located in peripheral sensory nerves, and their activation induces electrical pulses that are delivered to the brain and interpreted as 'pain.' Among such sensors, transient receptor potential ankyrin 1 (TRPA1), a polymodal cation channel, has a well-known molecular property of nociceptor and plays a pivotal role in various types of pain generation[1]. This is because TRPA1 is opened by a large number of irritants[2–4] and oxidative stimuli, such as reactive oxygen species (ROS)[5,6]. The reversible covalent or oxidative modification of cysteine or lysine residues on the N terminus of TRPA1 is necessary for channel opening in these situations[5–8]. We previously reported another mechanism for TRPA1 activation: hypoxia inhibits prolyl hydroxylase (PHD) activity, which relieves TRPA1 from the PHD-dependent hydroxylation of a proline residue located within the N-terminal ankyrin repeat domain (ARD) and thus leads to the opening of this channel[8]. More recently, we found that PHD inhibition is involved in the hindlimb ischaemia/reperfusion-evoked peripheral dysesthesia in mice[9].

Noxious thermal stimuli are danger signals for animals. The molecular basis governing how animals sense noxious hot stimuli has been well investigated, with some TRP[10,11] and other[12] channels identified as hot-sensing nociceptors. By contrast, the current understanding of how noxious cold stimuli are sensed shows limited progress, although the TRPM8-mediated mechanism for sensing innocuous cool stimuli is well defined[13,14]. TRPA1 was initially discovered to be a noxious cold-activated channel[2] and was a candidate cold nociceptor. However, the early literature shows some disagreement with subsequent studies about the role of TRPA1 as a cold sensor, and it is still debated whether TRPA1 is cold-sensitive[2,15–17] or not[3,4,18,19]. A recent study demonstrates that rodent but not human TRPA1 is cold-sensitive *in vitro*[20]. By contrast, a single-point mutation in the human TRPA1 (hTRPA1) gene is related to the familial episodic pain syndrome that is triggered by cold[21], and hTRPA1 is intrinsically a cold-gated channel[22].

Some chemotherapeutic agents induce peripheral neuropathy, including cold hypersensitivity, which leads to the discontinuation of the chemotherapy. The platinum-based chemotherapeutic agent oxaliplatin (L-OHP) often causes peculiar acute cold hypersensitivity, unlike other chemotherapeutic agents[23]. We previously reported that L-OHP, but not cisplatin or paclitaxel, elicits rapid-onset cold hypersensitivity in mice through the enhanced responsiveness of TRPA1 (refs 24,25). Furthermore, we found that the characteristic metabolite of L-OHP, oxalate, is responsible for the TRPA1-dependent cold hypersensitivity[24], but the molecular details remain unidentified.

In the present study, we hypothesize that the molecular basis of the PHD inhibition-mediated TRPA1 activation may also underlie the L-OHP-induced cold hypersensitivity, because the 1-carboxylate and 2-oxo functional groups of oxalate are common to a PHD cosubstrate, $\alpha$-ketoglutarate, and the artificial PHD inhibitor dimethyloxalylglycine (DMOG)[26]. Furthermore, since cold stimulation evokes mitochondria-derived ROS generation[27], we explore the relationship between the cold-dependent ROS and cold hypersensitivity. We show that the modification of a single proline residue on TRPA1 augments its sensitivity to $H_2O_2$, which endows TRPA1 with cold sensitivity via transduction of ROS signalling.

## Results

### Human TRPA1 exhibits cold sensitivity by a mutation of Pro[394].
We explored whether relief from PHD-mediated hydroxylation of the proline residue at position 394 (Pro[394]) within the N-terminal ARD of hTRPA1 could endow hTRPA1 with cold sensitivity in the presence of $H_2O_2$. In whole-cell recordings from HEK293 cells expressing wild-type hTRPA1 (hTRPA1-WT), cold stimulation (from 26 to 16 °C) did not affect whole-cell currents in the presence of a low concentration of $H_2O_2$ (0.1 µM), which had no effect on the current even at room temperature. By contrast, in mutant hTRPA1 lacking a hydroxylation-susceptible Pro[394] residue (hTRPA1-P394A), cold stimulation in the presence of $H_2O_2$-evoked whole-cell currents, which were significantly larger than those in hTRPA1-WT (Fig. 1a–d). The significant increase in the whole-cell currents of hTRPA1-P394A was observed only at 17 °C, not at 20 °C and 23 °C (Supplementary Fig. 1). Because $Ca^{2+}$ ions regulate the activity of TRPA1 (refs 18,28), we performed the same experiments using a $Ca^{2+}$-free bath solution (2 mM $CaCl_2$ was replaced with 1 mM EGTA). The hTRPA1-P394A was not activated by a combined stimulation of cold and $H_2O_2$ (Supplementary Fig. 2), suggesting that an enhanced effect of extracellular $Ca^{2+}$ ions[28] was also involved. Next, to determine whether cold directly activates hTRPA1-P394A, we performed inside-out patch-clamp recordings (Fig. 1e–g, Table 1). At 26 °C, hTRPA1-P394A showed higher open probability ($NP_O$) than did hTRPA1-WT, presumably owing to the higher basal activity of hTRPA1-P394A (ref. 8). In the absence of $H_2O_2$, cold stimulation (from 26 to 16 °C) failed to increase the $NP_O$ of hTRPA1-WT, but significantly increased the $NP_O$ of hTRPA1-P394A compared with the basal level (26 °C), although it did not reach statistical significance when compared with the $NP_O$ of hTRPA1-WT at 16 °C. By contrast, in the presence of 0.1 µM $H_2O_2$, the $NP_O$ of hTRPA1-P394A was significantly higher than that of hTRPA1-WT. Furthermore, 0.1 µM $H_2O_2$ increased the $NP_O$ of only hTRPA1-P394A at 16 °C, but not at 26 °C. Single-channel conductance of hTRPA1-P394A in the presence of 0.1 µM $H_2O_2$ at 16 °C was also significantly larger than the basal level (26 °C, no $H_2O_2$). These results suggest that hTRPA1-P394A may be directly activated by cold, but the presence of low concentrations of agonists is important for further activation and dilation[29] of hTRPA1-P394A, which cannot be achieved by hTRPA1-WT. This phenomenon may be due to the destabilization of the closed state of hTRPA1-P394A, but not hTRPA1-WT, at a low temperature, as previously described in rodents[30].

### Sensitization of hTRPA1 to ROS causes cold sensitivity.
We further investigated the cold sensitivity of hTRPA1 in intact cells using $Ca^{2+}$ imaging experiments. As expected, hTRPA1-WT-expressing cells showed little intracellular $Ca^{2+}$ concentration ($[Ca^{2+}]_i$) response to cold stimulation. By contrast, when hTRPA1-WT-expressing cells were pretreated with the PHD inhibitor DMOG (100 µM) for 2 h, the cold-evoked $[Ca^{2+}]_i$ responses were significantly facilitated, even in the absence of $H_2O_2$ (Fig. 2a,c). Similarly, hTRPA1-P394A-expressing cells showed significantly larger cold-evoked $[Ca^{2+}]_i$ responses than hTRPA1-WT-expressing cells (Fig. 2b,c). In HEK293 cells, we observed cold stimulation-induced $H_2O_2$ production, which was suppressed by pretreatment of a mitochondria-targeted anti-oxidant mitoTEMPO (10 µM; Supplementary Fig. 3), indicating that cold-evoked ROS is presumably derived from mitochondria, consistent with a previous report[27]. Thus, we examined the effects of mitoTEMPO on the cold-evoked $[Ca^{2+}]_i$ increase. Pretreatment with mitoTEMPO (10 µM) significantly suppressed the cold-evoked $[Ca^{2+}]_i$ increase in both DMOG-pretreated hTRPA1-WT- and hTRPA1-P394A-expressing cells (Fig. 2a–c). These results demonstrate that relieving hTRPA1 from prolyl hydroxylation enables the channel to sense cold by detecting cold-evoked ROS production.

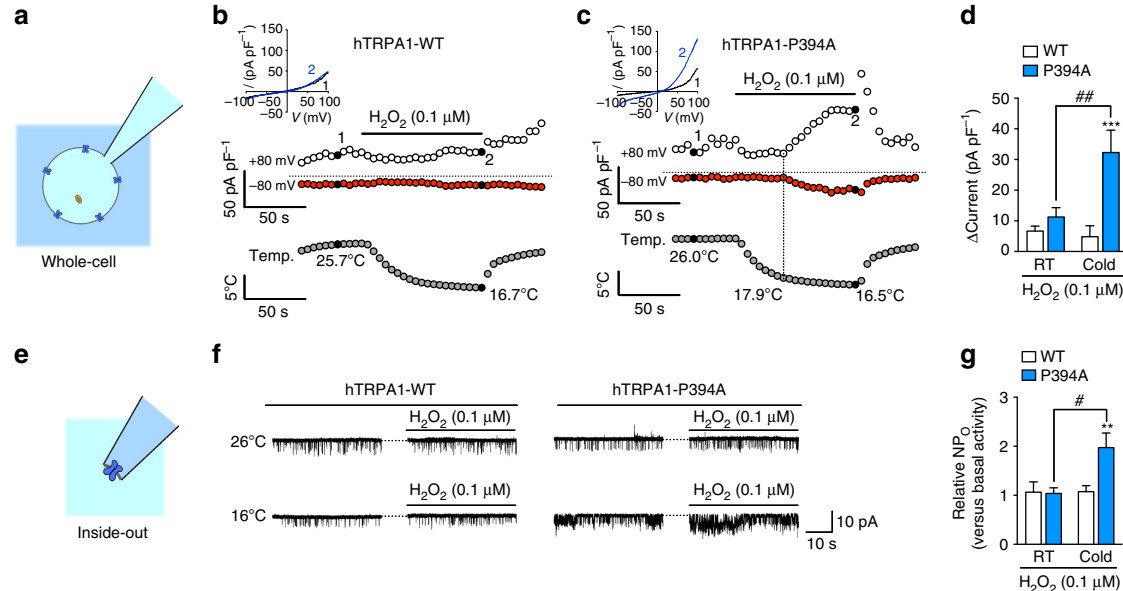

**Figure 1 | Hydroxylation-susceptible Pro394 in hTRPA1 is responsible for cold sensitivity in the presence of $H_2O_2$.** Either hTRPA1-WT- or hTRPA1-P394A-expressing cells were exposed to cold in the presence or absence of 0.1 μM $H_2O_2$ and analysed by whole-cell (**a–d**) or inside-out patch-clamp recordings (**e–g**). (**a–d**) Schematic diagram of a whole-cell patch-clamp configuration (**a**) along with representative whole-cell recordings from a hTRPA1-WT- (**b**) and a hTRPA1-P394A-expressing cell (**c**), and the statistical analysis (**d**, n = 7–8). Membrane potential was 0 mV. Insets in **b** and **c** are current–voltage relationships at the time indicated by black-filled circles. ***$P < 0.001$ versus hTRPA1-WT under the same condition; ##$P < 0.01$; two-way ANOVA with Sidak's *post hoc* test. (**e–g**) Schematic diagram of an inside-out patch-clamp configuration (**e**) along with representative single-channel currents of similar inside-out patches from a hTRPA1-WT- or hTRPA1-P394A-expressing cell in the presence or absence of 0.1 μM $H_2O_2$ at 26 °C (upper panels) or 16 °C (lower panels) (**f**), and a statistical analysis of the relative open probability ($NP_O$) compared with the basal $NP_O$ in RT or cold conditions (**g**, n = 13–14). Membrane potential was $-60$ mV. **$P < 0.01$ versus hTRPA1-WT under the same condition; #$P < 0.05$; two-way ANOVA with Sidak's *post hoc* test. All data are expressed as mean ± s.e.m. A detailed analysis of the inside-out patches is shown in Table 1. RT, room temperature.

**Table 1 | Parameters of hTRPA1-WT and hTRPA1-P394A activity in the cold and/or $H_2O_2$ stimulation.**

| | 26 °C | | 16 °C | |
| --- | --- | --- | --- | --- |
| | **Basal** | **$H_2O_2$ (0.1 μM)** | **Basal** | **$H_2O_2$ (0.1 μM)** |
| **hTRPA1-WT** | | | | |
| Conductance (pS) | 37.2 ± 1.7 | 34.2 ± 1.7 | 41.7 ± 2.9 | 43.6 ± 3.2 |
| $NP_O$ | 0.19 ± 0.07 | 0.15 ± 0.04 | 0.31 ± 0.08 | 0.30 ± 0.07 |
| **hTRPA1-P394A** | | | | |
| Conductance (pS) | 40.2 ± 1.8 | 42.2 ± 1.6 | 47.2 ± 3.5 | 51.2 ± 2.3** |
| $NP_O$ | 0.46 ± 0.19 | 0.61 ± 0.31 | 0.94 ± 0.36* | 1.43 ± 0.52***,† |

Single-channel conductance and open probability ($NP_O$) of single-channel currents were measured in hTRPA1-WT- or hTRPA1-P394A-expressing cells at 26 or 16 °C in the presence or absence of $H_2O_2$ (0.1 μM) using inside-out patches at a membrane potential of $-60$ mV as shown in Fig. 1e–g. n = 13–14.
*$P < 0.05$.
**$P < 0.01$.
***$P < 0.001$ versus data from the same channel at 26 °C in the absence of $H_2O_2$;
†$P < 0.01$ versus hTRPA1-WT under the same condition; two-way repeated measures ANOVA with Bonferroni's *post hoc* test. All data are expressed as mean ± s.e.m.

Because cold-evoked ROS is responsible for the pseudo-cold sensitivity of hTRPA1, we next examined whether inhibiting prolyl hydroxylation at room temperature solely causes the altered response of hTRPA1 to ROS. In hTRPA1-WT-expressing cells, the application of $H_2O_2$ (10 μM) evoked weak $[Ca^{2+}]_i$ responses. In DMOG-pretreated hTRPA1-WT-expressing cells (Fig. 2d,e) or hTRPA1-P394A-expressing cells (Fig. 2f,g), the $[Ca^{2+}]_i$ responses to $H_2O_2$ were significantly enhanced compared with those in hTRPA1-WT-expressing cells. We further compared the sensitivities of hTRPA1-P394A and hTRPA1-WT with other TRPA1 agonists (Supplementary Fig. 4). The $[Ca^{2+}]_i$ response of hTRPA1-P394A evoked by allyl isothiocyanate (AITC, 0.3 μM), a well-known TRPA1 agonist that activates TRPA1 via modification of cysteine residues[7], similar to ROS, was significantly higher than that of hTRPA1-WT. However, there

was no difference in the $[Ca^{2+}]_i$ responses to menthol (3 μM) and 2-aminoethoxydiphenyl borate (3 μM), TRPA1 agonists that activate TRPA1 via mechanisms independent of cysteine residues[7,31]. These results suggest that inhibiting prolyl hydroxylation enhances the sensitivity of hTRPA1 to a certain type of agonist, that is, agonists that modify the N-terminal cysteine residues of TRPA1.

**PHD inhibition induces cold hypersensitivity in mice.** To determine whether the phenomena that we observed for hTRPA1 also occur with mouse TRPA1 (mTRPA1), we performed the same experiments using mTRPA1-expressing cells (Fig. 3). In mTRPA1-expressing cells, cold stimulation evoked $[Ca^{2+}]_i$ responses (Fig. 3a,b), which were higher than those observed in

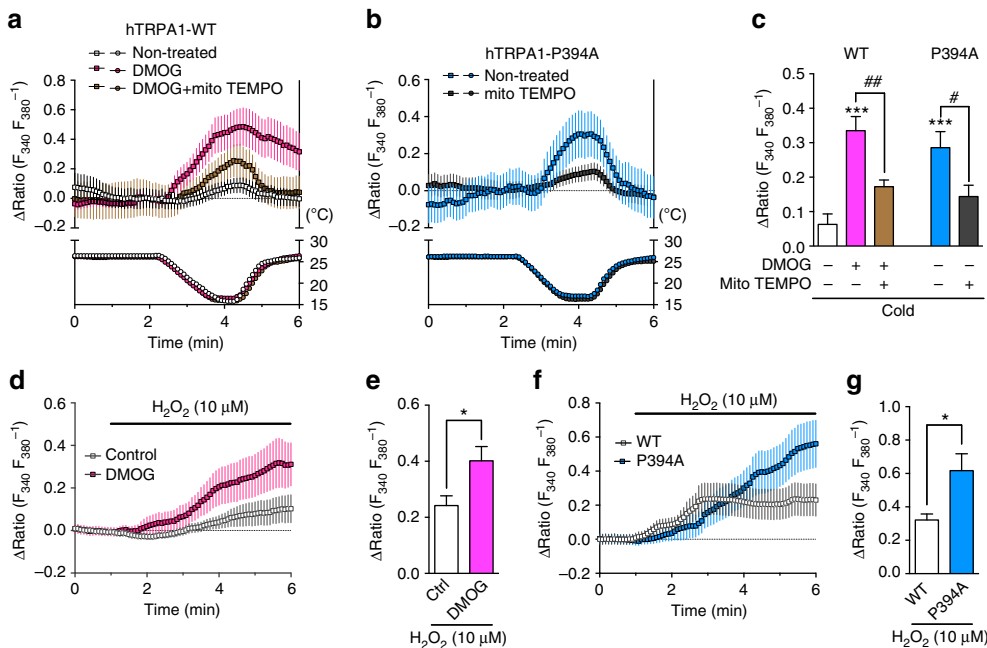

**Figure 2 | Enhanced response to ROS induced by relief from prolyl hydroxylation mediates cold sensitivity of hTRPA1.** DMOG-pretreated hTRPA1-WT-or hTRPA1-P394A-expressing cells were exposed to cold (**a–c**) or 10 μM $H_2O_2$ (**d–g**) and analysed using $Ca^{2+}$ imaging experiments. (**a–c**) Representative traces of cold-induced $[Ca^{2+}]_i$ responses in hTRPA1-WT-expressing cells pretreated with or without 100 μM DMOG for 2 h (**a**, $n = 7$–20 cells) and hTRPA1-P394A-expressing cells (**b**, $n = 11$–16 cells) with or without 10 μM mitoTEMPO pretreatment, and statistical analysis (**c**, $n = 45$–75 cells). ***$P < 0.001$ versus non-treated hTRPA1-WT (white bar); #$P < 0.05$, ##$P < 0.01$; one-way ANOVA with Tukey's *post hoc* test. (**d,e**) Representative traces of 10 μM $H_2O_2$-evoked $[Ca^{2+}]_i$ responses in hTRPA1-WT-expressing cells pretreated with vehicle (control) or 100 μM DMOG for 2 h (**d**, $n = 47$–55 cells), and statistical analysis (**e**, $n = 4$–8). *$P < 0.05$; unpaired $t$ test. (**f,g**) Representative traces of the $[Ca^{2+}]_i$ response evoked by 10 μM $H_2O_2$ in hTRPA1-WT-or hTRPA1-P394A-expressing cells (**f**, $n = 43$–46 cells), and statistical analysis (**g**, $n = 5$–6). *$P < 0.05$; unpaired $t$ test. All data are expressed as mean ± s.e.m. Ctrl, control.

hTRPA1-WT-expressing cells (see Fig. 2a,c). Consistent with hTRPA1, pretreatment with DMOG (100 μM) significantly enhanced cold-evoked $[Ca^{2+}]_i$ responses (Fig. 3a,b) and the 0.1 μM $H_2O_2$-induced $NP_O$ increase at 16 °C (Fig. 3c) in mTRPA1-expressing cells compared with those in non-treated mTRPA1-expressing cells. Similarly, the cold-evoked $[Ca^{2+}]_i$ response was significantly suppressed by the ROS scavenger N-*tert*-butyl-α-phenylnitrone (PBN, 3 mM; Fig. 3a,b). Furthermore, DMOG pretreatment significantly enhanced the 10 μM $H_2O_2$-evoked $[Ca^{2+}]_i$ response of mTRPA1 at room temperature (Fig. 3d).

Next, we performed cold-plate assays to determine whether our *in vitro* findings fit with an *in vivo* regulatory mechanism of TRPA1 in mice. Compared with the vehicle-pre-administration group, pre-administration of DMOG (400 mg kg$^{-1}$) significantly increased the cold escape behaviour. The DMOG-enhanced cold escape behaviours were significantly attenuated by the TRPA1-selective antagonist HC030031 (100 mg kg$^{-1}$) or PBN (100 mg kg$^{-1}$; Fig. 4a). These results suggest that PHD inhibition is sufficient to evoke cold hypersensitivity, which is dependent on TRPA1 and ROS signalling. We next verified whether ROS signalling also contributes to L-OHP-induced, rapid-onset cold hypersensitivity in mice. Consistent with our previous reports[24,25], pre-administration of L-OHP (5 mg kg$^{-1}$) significantly increased the cold escape behaviour. The L-OHP-enhanced cold escape behaviours were significantly attenuated by PBN (100 mg kg$^{-1}$; Fig. 4b). Taken together with the results of our previous study[24], our findings indicate that, similar to DMOG-induced cold hypersensitivity, L-OHP-induced cold hypersensitivity is dependent on TRPA1 and ROS signalling. To determine whether PHD inhibition can enhance the sensitivity

of TRPA1 to ROS *in vivo*, we examined the effect of DMOG on $H_2O_2$-evoked nocifensive behaviours in mice. Intraplantar (i.pl.) injection of $H_2O_2$ (0.5%, 20 μl) in mice evoked nocifensive behaviour, such as licking, flicking and biting of the injected paw. Pre-administration of DMOG (400 mg kg$^{-1}$) significantly increased the duration of the $H_2O_2$-evoked nocifensive behaviours, which were significantly attenuated by HC030031 (100 mg kg$^{-1}$; Fig. 4c). We next examined the effect of L-OHP on the $H_2O_2$-evoked nocifensive behavioural response. Because L-OHP is metabolized to oxalate and dichloro(1,2-diaminocyclohexane)platinum(II) (Pt(DACH)Cl$_2$), we performed similar experiments but using L-OHP as well as the membrane-permeable oxalate analogue dimethyl oxalate (DMO) and Pt(DACH)Cl$_2$. Pre-administration of L-OHP (5 mg kg$^{-1}$) or DMO (1.7 mg kg$^{-1}$) significantly enhanced $H_2O_2$-evoked nocifensive behaviours, and these effects of both compounds were significantly suppressed by HC030031 (100 mg kg$^{-1}$). However, pre-administration of Pt(DACH)Cl$_2$ (4.8 mg kg$^{-1}$) had no effect (Fig. 4d), suggesting that the enhanced TRPA1-mediated nocifensive behaviour induced by L-OHP is mediated through oxalate, but not its platinum metabolite.

We next performed *in vitro* experiments using cultured dorsal root ganglia (DRG) neurons derived from wild-type or TRPA1-knockout (KO) mice. In wild-type DRG neurons, applying $H_2O_2$ (100 μM) induced no or weak $[Ca^{2+}]_i$ responses. When wild-type DRG neurons were pretreated with L-OHP (100 μM) or DMO (30 μM) for 2 h, the $[Ca^{2+}]_i$ responses to $H_2O_2$ at room temperature were significantly augmented compared with those of vehicle-pretreated DRG neurons. By contrast, few TRPA1-KO-DRG neurons responded to $H_2O_2$ even following pretreatment with L-OHP or DMO (Fig. 5), indicating that the agents

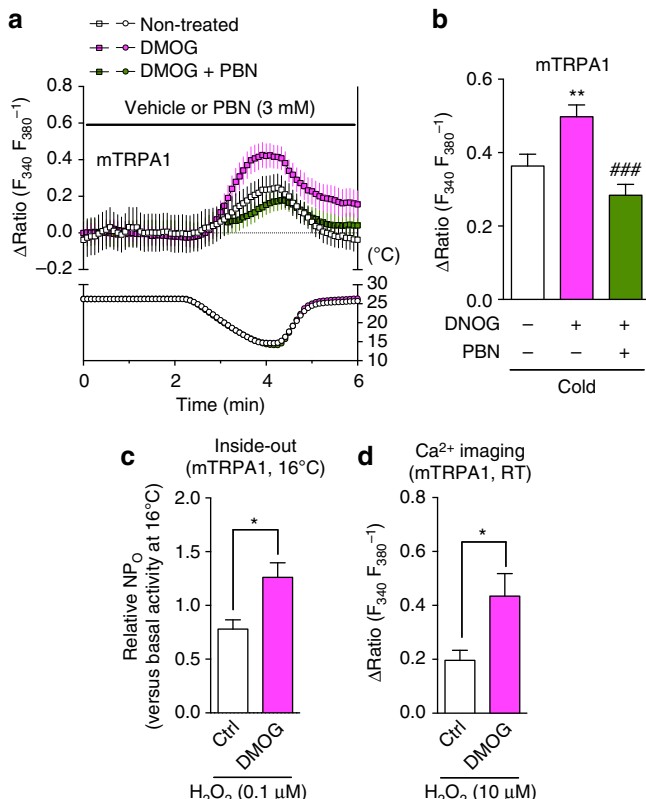

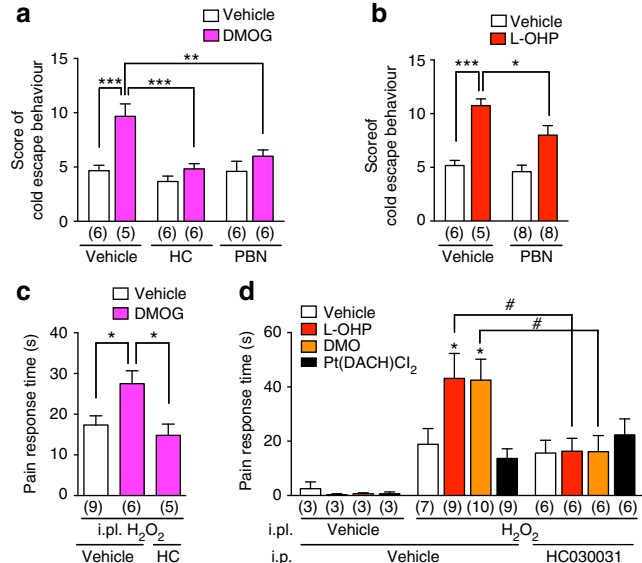

**Figure 3 | PHD inhibition augments cold-induced activation of mTRPA1 via an enhanced response to ROS.** mTRPA1-expressing cells were exposed to cold (**a–c**) or 10 μM $H_2O_2$ (**d**) and analysed using $Ca^{2+}$ imaging experiments or inside-out patch-clamp recordings. (**a**) Representative traces of cold-induced $[Ca^{2+}]_i$ responses in mTRPA1-expressing cells pretreated with or without DMOG (100 μM) for 2 h (n = 8–19 cells) in the presence or absence of 3 mM PBN, and (**b**) statistical analysis (n = 39–96 cells). **P < 0.01 versus non-treated mTRPA1 (white bar); ###P < 0.001 versus DMOG-pretreated mTRPA1 (magenta bar); one-way ANOVA with Tukey's *post hoc* test. (**c**) Relative open probability (NP_O) of mTRPA1 in inside-out patches in the presence 0.1 μM $H_2O_2$ at 16 °C from a mTRPA1-expressing cell pretreated with or without DMOG (100 μM) for 2 h. n = 8–11. *P < 0.05; unpaired *t* test. Membrane potential was −60 mV. (**d**) Statistical analysis of the $H_2O_2$-evoked $[Ca^{2+}]_i$ responses in mTRPA1-expressing cells pretreated with or without DMOG (100 μM) for 2 h. n = 10–12. *P < 0.05; unpaired *t* test. All data are expressed as mean ± s.e.m.

**Figure 4 | Inhibiting PHDs contributes to TRPA1-mediated cold hypersensitivity via an enhanced response to ROS.** Mice were pre-administered DMOG or L-OHP, and their cold escape behaviour in the cold-plate test (**a,b**) or nocifensive behaviours evoked by intraplantar injection (i.pl.) of $H_2O_2$ (**c,d**) were examined. (**a,b**) Effects of DMOG (**a**, 400 mg kg$^{-1}$, i.p.) or L-OHP (**b**, 5 mg kg$^{-1}$, i.p.) pre-administration with or without HC030031 (HC; 100 mg kg$^{-1}$, i.p.) or PBN (100 mg kg$^{-1}$, i.p.) on cold escape behaviours. Numbers in parentheses indicate the number of animals in each group. *P < 0.05; **P < 0.01; ***P < 0.001; two-way ANOVA with Sidak's *post hoc* test. (**c**) Effects of DMOG pre-administration with or without HC030031 on $H_2O_2$-evoked nocifensive behaviours. *P < 0.05; one-way ANOVA with Tukey's *post hoc* test. (**d**) Effects of L-OHP, DMO (1.7 mg kg$^{-1}$) or Pt(DACH)Cl$_2$ (4.8 mg kg$^{-1}$) pre-administration with or without HC030031 on $H_2O_2$-evoked nocifensive behaviours. *P < 0.05 versus i.pl. $H_2O_2$- and i.p. vehicle-treated vehicle group (white bar); #P < 0.05; two-way ANOVA with Tukey's *post hoc* test. All data are expressed as mean ± s.e.m.

selectively enhance the sensitivity of TRPA1, but not other ROS-sensitive channels, in DRG neurons.

**L-OHP induces hTRPA1 sensitization via PHD inhibition.** We further examined the molecular basis underlying the L-OHP-induced hTRPA1 sensitization. In hTRPA1-WT-expressing HEK293 cells, pretreatment with L-OHP (100 μM) or DMO (30 μM) for 2 h significantly enhanced the $[Ca^{2+}]_i$ responses to $H_2O_2$ (10 μM), whereas pretreatment with Pt(DACH)Cl$_2$ (30 μM) had no effect (Supplementary Fig. 5). The L-OHP-enhanced $H_2O_2$-evoked $[Ca^{2+}]_i$ increase was not affected by co-pretreatment with the antioxidants glutathione (1 mM) or PBN (10 mM; Supplementary Fig. 5a), indicating that the ROS presumably generated during L-OHP pretreatment[32] was dispensable for the induction of TRPA1 sensitization.

To determine whether the L-OHP- or DMO-induced hTRPA1 sensitization to ROS is PHD-dependent, we tested the effects of

L-OHP and DMO on the amount of hypoxia-inducible factor-1α (HIF-1α), whose degradation is principally regulated by PHDs[33]. Treatment of HEK293 cells with L-OHP (100 μM) or DMO (30 μM) for 24 h increased the amount of HIF-1α similar to that induced by CoCl$_2$ (100 μM), a pan inhibitor of PHDs (Fig. 6a), suggesting that L-OHP and DMO have the ability to inhibit PHDs. Furthermore, pretreatment with L-OHP (100 μM) or DMO (30 μM) for 2 h failed to further augment the $H_2O_2$-evoked $[Ca^{2+}]_i$ increase in hTRPA1-P394A-expressing cells (Fig. 6b) or cells overexpressing a catalytically dead mutant of human PHD 2 (mutPHD2) in hTRPA1-WT-expressing cells (Fig. 6c). To compete with the inhibitory effect of L-OHP on PHDs, we overexpressed recombinant human PHD2 (wtPHD2) in the hTRPA1-expressing cells. Overexpression of wtPHD2 abolished the L-OHP-induced augmentation without affecting the basal $H_2O_2$-evoked $[Ca^{2+}]_i$ increase (Fig. 6c), which may be due to incomplete inhibition of endogenous and exogenous PHDs by L-OHP. These results demonstrate that L-OHP or oxalate increases the sensitivity of hTRPA1 to ROS through inhibition of the PHD-mediated hydroxylation of Pro$^{394}$ in hTRPA1.

**Discussion**

In the present study, we found that relief from the PHD-mediated hydroxylation of a proline residue in TRPA1 augments its sensitivity to ROS and endows TRPA1 with cold sensitivity,

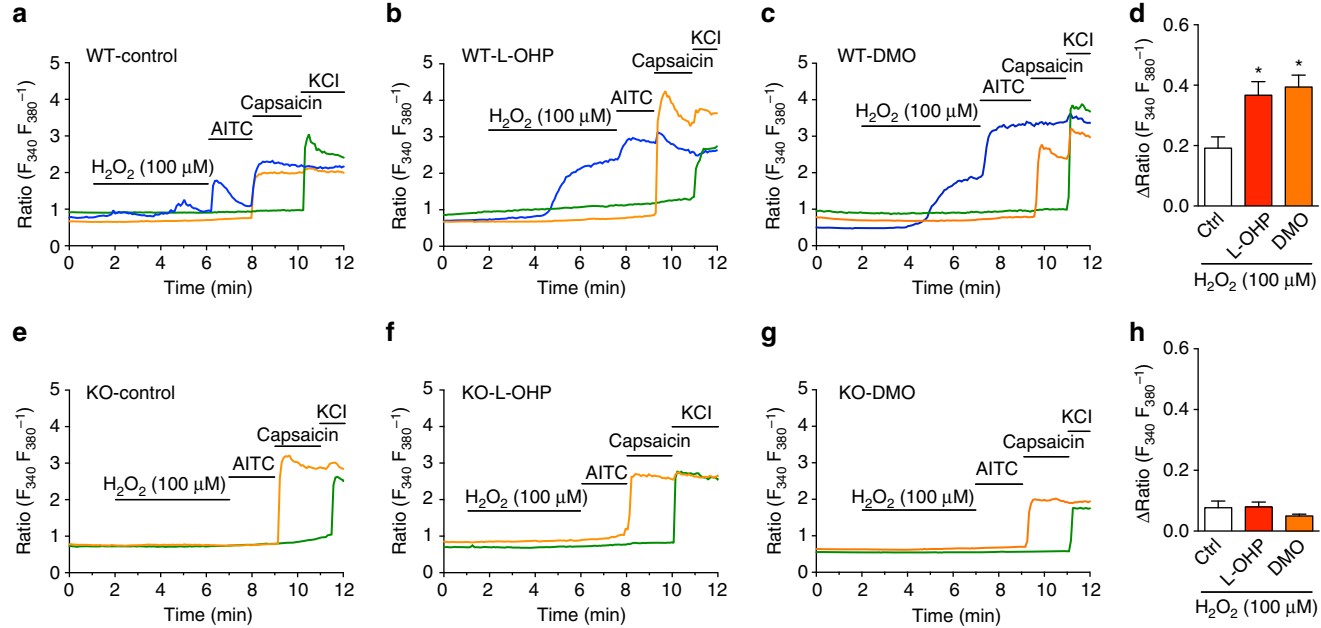

**Figure 5 | L-OHP or DMO augments the $H_2O_2$-evoked $[Ca^{2+}]_i$ increase in mouse DRG neurons.** Shown are representative traces of 100 μM $H_2O_2$-induced $[Ca^{2+}]_i$ responses in DRG neurons derived from wild-type (WT, **a–c**) or TRPA1-KO (**e–g**) mice pretreated with vehicle (**a,e**; control), 100 μM L-OHP (**b,f**) or 30 μM DMO (**c,g**) for 2 h. Statistical analysis of the $H_2O_2$-evoked $[Ca^{2+}]_i$ increase is shown in **d** (WT-DRG; $n = 5$–6) and **h** (TRPA1-KO-DRG; $n = 4$–5). *$P < 0.05$ versus each control (Ctrl); one-way ANOVA with Tukey's *post hoc* test. AITC, 100 μM; capsaicin, 0.3 μM; KCl, 50 mM. No difference was detected in the AITC-evoked $[Ca^{2+}]_i$ increase among groups of WT-DRG neurons (relative ΔRatio versus vehicle: L-OHP, $1.02 \pm 0.06$; DMO, $1.14 \pm 0.13$). All data are expressed as mean ± s.e.m.

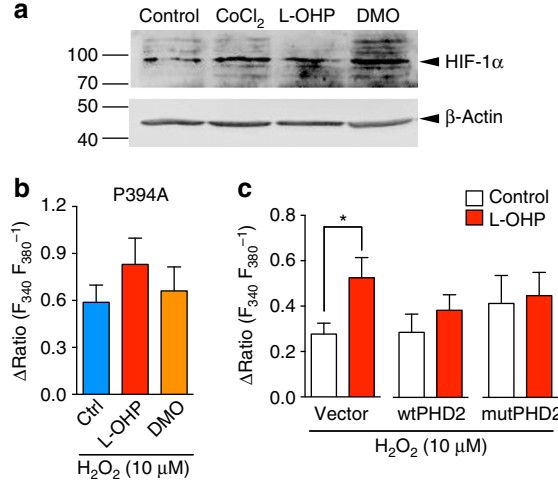

**Figure 6 | L-OHP and DMO elicit hTRPA1 sensitization via PHD inhibition.** (**a**) Representative western blots of whole-cell lysates of HEK293 cells pretreated with vehicle, 100 μM $CoCl_2$, 100 μM L-OHP or 30 μM DMO for 24 h. $CoCl_2$ (100 μM) was used as a positive control. (**b,c**) Statistical analysis of the $H_2O_2$-evoked $[Ca^{2+}]_i$ increase in hTRPA1-P394A-expressing cells (**b**, $n = 5$–6) or hTRPA1-WT-expressing cells cotransfected with vector, hPHD2 (wtPHD2) or mutant hPHD2 (mutPHD2) (**c**, $n = 5$–7) pretreated with or without L-OHP (100 μM) or DMO (30 μM). *$P < 0.05$; unpaired *t*-test. All data are expressed as mean ± s.e.m.

which is one mechanism underlying L-OHP-induced acute cold hypersensitivity (Fig. 7). To the best of our knowledge, this is the first report indicating that chemical modification allows genetically intact hTRPA1 to be activated by cold via sensing cold-induced ROS generation.

ROS, such as superoxide anion and $H_2O_2$, are classically accepted as toxic substances. However, recent studies revealed that non-toxic levels of ROS have an alternative role as a signalling molecule. Some TRP channel members contribute to this signalling[34]. Among them, TRPA1 is the principal candidate that regulates ROS signalling because it is the most redox-sensitive TRP channel[8]. ROS-induced TRPA1 activation plays a critical role in airway hypersensitivity[35] and cerebral artery dilation[36]. In addition, in peripheral sensory nerves, endogenously generated redox molecules, such as $H_2O_2$ (ref. 37) and nitroxyl[38], are able to regulate TRPA1. These observations suggest that TRPA1 acts as a polymodal sensor for detecting ROS induced by multiple environmental changes. In our study, noxious cold-induced $H_2O_2$ (presumably from mitochondria) was determined to be the key molecule regulating TRPA1.

Whether TRPA1 is cold-sensitive remains a matter of debate. TRPA1 was first identified as a sensor of noxious cold ($\leq 17\,^\circ$C)[2], and subsequent studies showed the role of TRPA1 in noxious cold sensation *in vitro* and *in vivo*[15–17]. Furthermore, a point mutation (N855S) in the S4 transmembrane segment of TRPA1 was identified as the cause of autosomal-dominant familial episodic pain syndrome characterized by episodes of debilitating upper body pain that is triggered by fasting and physical stress, including noxious cold[21], suggesting that hTRPA1 may also be associated with noxious cold sensation. However, this idea is challenged by other studies showing that TRPA1 is not activated directly by cold and fails to contribute to cold nociception[3,4,18,19]. A recent study concludes that the cold activation of TRPA1 is species specific: TRPA1 in mouse and rat is activated by cold, but TRPA1 in human and rhesus monkey is cold-insensitive[20]. However, hTRPA1 purified and reconstituted into lipid bilayers is cold-sensitive both in the presence and absence of the N-terminal ARD domain[22]. This study contradicts other studies indicating that hTRPA1 is generally cold-insensitive[4,18,20]. Furthermore, an electrophilic TRPA1 agonist AITC is able to activate purified hTRPA1 lacking the N-terminal ARD domain[22], whereas another report also indicated that covalent modification of the three cysteine residues within the N-terminal ARD domain is necessary

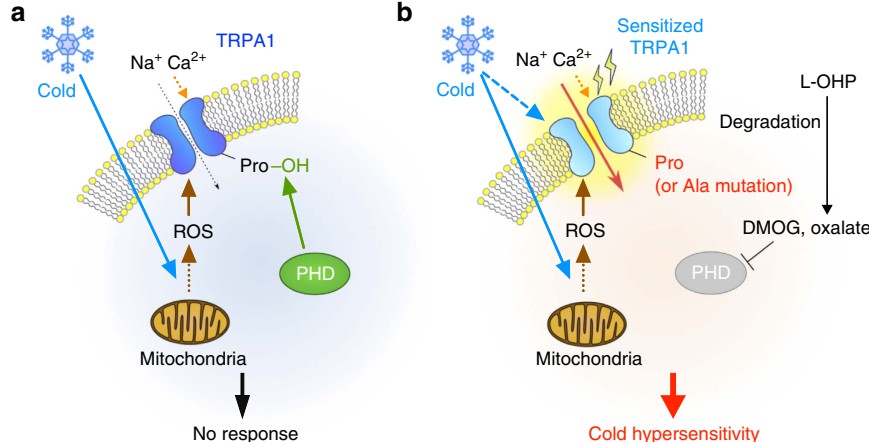

**Figure 7 | Consequence of PHD inhibition-triggered TRPA1-mediated cold hypersensitivity.** (**a**) Cold triggers ROS generation, but cold is insufficient to activate TRPA1 with conserved PHD-mediated hydroxylation of a proline residue. (**b**) Inhibition of PHD (or mutating the proline residue to alanine) enhances the sensitivity of TRPA1 to ROS. This enhanced sensitivity enables TRPA1 to sense cold-evoked ROS, which causes cold hypersensitivity. The dotted arrows indicate the indirect or multiple pathway.

for AITC-induced hTRPA1 activation[7]. These discrepancies may be the result of experimental conditions; a purified channel protein in a defined membrane environment does not interact with any other molecules. By contrast, a non-purified channel in a living cell can interact with other regulatory molecules, such as annexin A2 (ref. 39), TRPA1b (ref. 19), TRPV1 and Tmem100 (ref. 40). In the present study, although we found that mTRPA1 was intrinsically cold-sensitive compared with hTRPA1 (compare Fig. 2 and Fig. 3), DMOG pretreatment further augmented the cold-induced mTRPA1 activation, and this was suppressed by a ROS scavenger both *in vitro* and *in vivo*. These results indicate that ROS is important for the cold-evoked mTRPA1 activation followed by PHD inhibition-induced cold hypersensitivity, although another mechanism underlies the intrinsic cold sensitivity of mTRPA1. However, we could not fully exclude the possibility that the basal level of $H_2O_2$, in addition to the cold-evoked $H_2O_2$ production, also contributes to the cold hypersensitivity of TRPA1. Furthermore, although we consider that the extracellular $Ca^{2+}$ ion modulates TRPA1 directly, it is possible that $Ca^{2+}$ influx triggers ROS production from mitochondria[41] thereby contributes to TRPA1 activation. Nevertheless, our findings are unique in that the sensitized but cold-insensitive hTRPA1 is actually able to sense noxious cold with the aid of ROS.

L-OHP induces acute cold hypersensitivity in ~90% of patients during or within hours after infusion[23,42]. In addition to our previous findings[24,25], the present results further support the possibility that the enhanced sensitivity of TRPA1 to ROS via PHD inhibition underlies the cold hypersensitivity induced by L-OHP, although we cannot ignore the contribution of TRPM8 (refs 43,44) or $Na_V1.6$ (refs 45,46). Since DMOG, which does not induce ROS generation, is sufficient to induce ROS- and TRPA1-dependent cold hypersensitivity in mice, oxalate, a characteristic metabolite of L-OHP, seems responsible for the acute cold hypersensitivity[24,45–48] via PHD inhibition. The molecular mechanism underlying hypoxia- and PHD-mediated sensitization of sensory TRPA1 to cold and ROS may also underlie the pathophysiological process of cold hypersensitivity induced by peripheral ischaemia, such as in Buerger's disease[49] and Raynaud syndrome[50].

Recent ongoing studies mainly focused on the intrinsic thermosensitivity of ion channels[51]. Our study represents a significant paradigm shift; TRPA1 is able to act as a noxious cold sensor by transducing cold-induced ROS signalling into neuronal activity, without directly sensing noxious cold. The molecular mechanisms underpinning how noxious cold under physiological conditions triggers neuronal excitation are still unclear, although $Na_V1.8$ (ref. 52) and $Na_V1.9$ (ref. 53) have been implicated in these processes. Although we could not clarify the molecular mechanism how cold triggers mitochondrial ROS generation, it is reported that mitochondrial ROS generated during cold stimulation has a pivotal role in thermogeneration[54]. Thus, the physiological importance of TRPA1-mediated cold detection via ROS signalling warrants further investigation.

## Methods

**Animals.** The C57BL/6J mice 6–8 weeks old were purchased from Japan SLC (Shizuoka, Japan). The $Trpa1^{-/-}$ mouse line (TRPA1-KO mice), obtained from Jackson Laboratory (Bar Harbor, ME), was backcrossed to C57BL/6J mice for at least ten generations and genotyped by genomic PCR using the following primers: 5′-tcatctgggcaacaatgtcacctgct-3′ and 5′-tcctgcaagggtgattgcgttgtcta-3′. All mice were housed in constant ambient temperature ($24 \pm 1\,^{\circ}C$) and humidity ($55\% \pm 10\%$) conditions and under a 12 h light–dark cycle. The mice ate chow and drank water *ad libitum*. All animal care and experimental procedures were performed in accordance with the ethical guidelines of the Kyoto University Animal Research Committee. The protocol was approved by the Kyoto University Animal Research Committee (permission number: 2014–45, 2015–38).

**Reagents.** Oxaliplatin (L-OHP), $H_2O_2$ and AITC were purchased from Wako Pure Chemical Industries (Osaka, Japan). HC030031 was obtained from Shanghai Haoyuan Chemexpress (Shanghai, China). Menthol, 2-aminoethoxydiphenyl borate, PBN, DMO, Pt(DACH)Cl$_2$, cremophor EL, poly-L-lysine, D-mannitol, glycerol, 2,2′,4,4′-tetrahydroxybenzophenone, magnesium turnings, hexane, DMF, THF, iodine and ethyl acetate were purchased from Sigma-Aldrich (St Louis, MO). The BAPTA (1,2-bis(2-aminophenoxy)ethane-N,N,N′,N′-tetraacetic acid) was acquired from Dojindo Laboratories (Kumamoto, Japan). DMOG was purchased from Frontier Scientific (Logan, Utah). The mitoTEMPO was obtained from Santa Cruz (Dallas, TX). Laminin was acquired from Life Technologies (Carlsbad, CA). N,N-diisopropylethylamine, 2-bromo-5-methoxytoluene, TBDMS-Cl and PdCl$_2$(dppf)·CH$_2$Cl$_2$ complex (1:1) were purchased from Tokyo Chemical Industry (Tokyo, Japan). Sodium acetate and bis(pinacolato)diboron was purchased from WAKO (Osaka, Japan). Other drugs and chemicals were obtained from Nacalai Tesuque (Kyoto, Japan).

**Synthesis of PG-1.** PG-1 was synthesized according to the literature[55,56] with slight modifications. The details are described as follows. NMR spectra were collected in DMSO-d6 or CDCl$_3$ at 23–26 °C on JNM-ECZ500R (JEOL) at Technical Support Office, Department of Synthetic Chemistry and Biological Chemistry, Graduate School of Engineering, Kyoto University. All chemical shifts are reported in the standard δ notation of parts per million using TMS as reference. Mass spectra were measured on an Exactive (Thermo Fisher Scientific) equipped

with electron spray ionization (ESI) at Technical Support Office, Department of Synthetic Chemistry and Biological Chemistry, Graduate School of Engineering, Kyoto University.

3,6-dihydroxyxanthone (**1**). 2,2′,4,4′-Tetrahydroxybenzophenone (7.6 g, 31 mmol) and sodium acetate (25 g, 305 mmol) were dissolved in 175 ml of water. This mixture was refluxed at 115 °C for 20 h under air. This was transferred to 0 °C and 1 N HCl (aq.) was added to bring the solution to around pH 4–5. The crystal was collected by filtration and vacuum dried to afford product **1** (6.8 g, 97% yield). $^1$H-NMR (DMSO-d6, 500 MHz): δ 10.81 (2H, s), 7.98 (2H, d), 6.86 (2H, dd), 6.82 (2H, d).

3,6-Bis-(tert-butyldimethylsilanyloxy)xanthen-9-one (**2**). Imidazole (1.65 g, 22.5 mmol) and **1** (1.25 g, 5.5 mmol) were dissolved in 7.5 ml dry DMF under $N_2$. TBDMS-Cl (2.48 g, 16 mmol) dissolved in 5 ml dry DMF was added to this mixture. After stirring at room temperature for 2 h, the mixture was diluted with ∼20 ml hexane/ethyl acetate (1:1). The organic layer was washed with water, 0.1% NaOH, brine and the solvent was removed by rotary evaporation to afford product **2** (2.1 g, 82% yield). $^1$H-NMR (CDCl$_3$, 500 MHz): δ 8.20 (2H, dd), 6.86 (2H, d), 6.84 (2H, d), 1.01 (18H, s) and 0.29 (12H, s).

9-[1-(2-Methyl-4-methoxyphenyl)]-6-hydroxy-3H-xanthen-3-one (**3**). Magnesium turnings (0.16 g, 6.6 mmol) and iodine (1 pellet) were dissolved in 7.5 ml of anhydrous THF under $N_2$. To this mixture, 2-bromo-5-methoxytoluene (1.6 g, 8.0 mmol) dissolved in 7.5 ml anhydrous THF was added. This was refluxed at 60 °C for 1 h. The solution was cooled to room temperature, and was added to a mixture of **2** (1.2 g, 2.6 mmol) in 15 ml of anhydrous THF pre-cooled to −70 °C. This mixture was transferred to 0 °C and was stirred for 10 min. This mixture was brought back to room temperature and stirred for additional 3 h under $N_2$. 3 ml of 4 N HCl(aq) was added and further refluxed at 60 °C for 2 h. 5 ml of 4 N NaOH followed by saturated NaH$_2$PO$_4$ was added to bring the solution to around pH 4. The solvent was removed by rotary evaporation and the resulting red precipitate was collected by filtration. This was resuspended in methanol, refluxed at 90 °C for 1 h to dissolve completely and was crystalized at −30 °C overnight The crystal was collected by filtration to afford product **3** (680 mg, 79% yield). ESI–MS calculated for [MH$^+$] 333.1121, found 333.1116.

9-[1-(2-Methyl-4-methoxyphenyl)]-6-trifluoromethanesulfonate-3H-xanthen-3-one (**4**). N-phenyl bis(trifluoromethanesulfonamide; 472 mg, 1.3 mmol) and **3** (400 mg, 1.2 mmol) were dissolved in 20 ml of dry DMF. N,N-diisopropyl-ethylamine (Hünig's base, 0.60 ml, 3.5 mmol) was added via syringe, and the resulting solution was stirred at room temperature in the dark under $N_2$ overnight. The reaction was concentrated by rotary evaporation and diluted with ∼100 ml ethyl acetate. The organic layer was washed twice with 5% citric acid, twice with water, once with brine and dried over Na$_2$SO. The solvent was removed by rotary evaporation. Purification by silica gel column chromatography with hexane/ethyl acetate (1:1) afforded product **4** as yellow–orange crystals (280 mg, 50% yield). $^1$H-NMR (CDCl$_3$, 500 MHz): δ 7.39 (1H, s), 7.18 (1H, d), 7.08 (2H, m), 7.02 (1H, d), 6.94 (2H, m), 6.59 (1H, dd), 6.43 (1H, d), 3.90 (3H, s), 2.06 (3H, s). ESI–MS calculated for [MH$^+$] 465.0614, found 465.0609.

9-[1-(2-Methyl-4-methoxyphenyl)]-6-pinacolatoboron-3H-xanthen-3-one (Peroxy Green 1, PG-1 (**5**). Bis(pinacolato)diboron (60 mg, 0.24 mmol) and PdCl$_2$(dppf)·CH$_2$Cl$_2$ complex (1:1) (60 mg, 0.073 mmol) were added to a flask under Ar. A suspension of potassium acetate (128 mg, 1.33 mmol) and **4** (200 mg, 0.44 mmol) in anhydrous 1,4-dioxane (24 ml) was added to this mixture. This mixture was refluxed at 100 °C for 8 h under nitrogen. The reaction was cooled to room temperature and diluted with pentane (25 ml). The organic layer was washed with brine (4 × 25 ml), dried over Na$_2$SO$_4$, and the solvent was removed by rotary evaporation. Purification by silica gel column chromatography with hexane/ethyl acetate (1:3) afforded product **PG-1** (**5**) as an orange solid (12.3 mg, 6.3% yield). $^1$H-NMR (CDCl$_3$, 500 MHz): δ 7.90 (1H, s), 7.56 (1H, d), 7.08 (2H, d), 7.02 (1H, d), 6.92 (2H, m), 6.59 (1H, dd), 6.44 (1H, d), 3.91 (3H, s), 2.04 (3H, s) and 1.36 (12H, s). $^{13}$C NMR (125.65 MHz, CDCl$_3$) δ 186.2, 160.5, 158.8, 151.9, 148.8, 138.0, 131.1, 130.7, 130.5, 129.9, 127.2, 124.5, 123.1, 122.9, 122.0, 116.1, 111.6, 105.9, 84.5, 55.4, 29.7, 24.9 and 20.0. ESI–MS calculated for [MH$^+$] 443.2024, found 443.2022.

**Plasmids.** Mouse TRPA1 (GenBank accession No. NM177781.4) was cloned from mouse brain, whole Marathon-Ready cDNA (BD Biosciences, Franklin Lakes, NJ) by applying a PCR-based approach designed to contain the consensus sequence from the translation initiation, and was subcloned into the expression vector pCI-neo (Promega Corporation, Madison, WI) using XhoI and SalI sites. The primer sequences used for the cloning of mouse TRPA1 are summarized in Supplementary Table 1. As the hTRPA1 cDNA, hTRPA1-P394A mutant cDNA, recombinant hPHD2 cDNA, and hPHD2 mutant cDNA in the pCIneo expression vector, we used the same plasmids previously constructed[8]. The pEGFP-C3 was purchased from Clontech Laboratories (Madison, WI).

**Cell line cultures and cDNA transfection.** HEK293 cells were cultured in DMEM with GlutaMAX I (10566-016, Life Technologies) supplemented with 10% heat-inactivated fetal bovine serum (Sigma) and maintained at 37 °C in a humidified incubator set at 5% CO$_2$. HEK293 cells were cotransfected with recombinant plasmids and pEGFP-C3 using SuperFect Transfection Reagent (Qiagen, Hilden, Germany) or Lipofectamine 2000 (Life Technologies). Two days after transfection,

cells were placed onto coverslips coated with poly-L-lysine and used in electro-physiological recording or fluorometric imaging.

**Primary cultures of mouse dorsal root ganglion neurons.** Bilateral L1–L6 dorsal root ganglions (DRGs) were collected from a freshly killed adult C57BL/6J male or female mouse (6–8 weeks old). DRGs were incubated for 1 h at 37 °C in Hank's balanced salt solution containing 137 mM NaCl, 5.4 mM KCl, 0.34 mM Na$_2$HPO$_4$, 0.44 mM KH$_2$PO$_4$, 5.6 mM D-glucose and 2.4 mM HEPES (adjusted to pH 7.4 with NaOH), which contained 0.3% collagenase and 0.4% dispase. A Percoll (Sigma-Aldrich) gradient was used to separate DRG neurons from myelin and nerve debris as follows. Solutions of 30 and 60% Percoll were prepared with L15 medium. The 30% Percoll solution was gently layered over the 60% Percoll solution, and the cell suspension was gently layered over the Percoll gradient. After 10 min of centrifugation at 1,800g, the cells were collected from the Percoll interface, suspended in 8 ml of L15 medium, and centrifuged again for 5 min at 1,800g. The supernatant was removed, and the cell pellet was resuspended in 70 μl of DMEM (D6046, Sigma) containing 10% heat-inactivated fetal bovine serum (Sigma), penicillin G (100 U ml$^{-1}$) and streptomycin (100 μg ml$^{-1}$). The cells were plated onto laminin-coated coverslips and incubated at 37 °C in a humidified 5% CO$_2$ atmosphere. After 4 h of incubation, 1.5 ml of DMEM was added and the cells were incubated again, but this time overnight at 37 °C in a humidified 5% CO$_2$ atmosphere.

**Electrophysiology.** Electrophysiological recordings were performed with a pipette made from a glass capillary (outer diameter, 1.5 mm) with a filament inside (Narishige, Tokyo, Japan) pulled using a P-87 micropipette puller (Sutter, Novato, CA). The access resistance ranged 2–5 MΩ when the pipette was filled with the pipette solution described below. For whole-cell patch-clamp recordings, the bath solution contained 100 mM NaCl, 2 mM CaCl$_2$ and 10 mM HEPES (adjusted to pH 7.4 with NaOH and 300 mOsm with D-mannitol), and the pipette solution contained 100 mM Cs-aspartate, 5 mM BAPTA, 1.4 mM Ca-gluconate (30 nM free Ca$^{2+}$), 2 mM MgSO$_4$, 2 mM MgCl$_2$, 4 mM Na$_2$-ATP, 10 mM Na$_5$P$_3$O$_{10}$ and 10 mM HEPES (adjusted pH 7.4 with CsOH and 300 mOsm with D-mannitol). In the Ca$^{2+}$-free bath solution, CaCl$_2$ was omitted and 1 mM EGTA was added. Current–voltage relationships were measured using voltage ramps (−100 to +100 mV over 100 ms) applied every 5 s. The membrane potential was set at 0 mV. Access resistances were compensated by 70%. Data were filtered at 2.9 kHz. For inside-out patch-clamp recordings, the bath solution contained 50 mM Cs-aspartate, 50 mM CsCl, 10 mM EGTA, 1 mM CaCl$_2$ (10 nM free Ca$^{2+}$), 1 mM MgCl$_2$, 10 mM Na$_5$P$_3$O$_{10}$ and 10 mM HEPES (adjusted to pH 7.4 with CsOH and 300 mOsm with D-mannitol), and the pipette solution contained 100 mM CsCl, 1 mM MgCl$_2$, 1 mM EGTA and 10 mM HEPES (adjusted to pH 7.4 with CsOH and 300 mOsm with D-mannitol). The membrane potential was set at −60 mV. Data were filtered at 2.0 kHz. Experiments were performed at room temperature unless otherwise stated. Cold stimulation was performed with an SC-20 dual in-line solution heater/cooler and a CL-100 temperature controller (Warner Instruments, Hamden, CT). Patch-clamp recordings were performed using an EPC-10 patch-clamp amplifier (HEKA Instruments, Lambrecht, Germany) and PATCHMASTER software (HEKA), and analysed using IGOR Pro software (Wave Metrics, Portland, OR) and Clampfit software (Molecular Devices, Sunnyvale, CA).

**Measurement of intracellular Ca$^{2+}$ concentration ([Ca$^{2+}$]$_i$).** Cells on cover-slips were loaded for 30–40 min with 5 μM Fura-2 acetoxymethyl ester (Fura-2 AM; Dojindo Laboratories) in Krebs–Ringer solution containing 140 mM NaCl, 5 mM KCl, 1 mM MgCl$_2$, 2 mM CaCl$_2$, 10 mM glucose and 10 mM HEPES, which contained 0.005% cremophore EL. Fluorescence images were captured every 5 s using alternating excitation at 340 and 380 nm and emission at 510 nm with an AQUACOSMOS/ORCA-AG imaging system (Hamamatsu Photonics, Shizuoka, Japan). For pretreatment, DMOG (100 μM), L-OHP (100 μM), DMO (30 μM) or Pt(DACH)Cl$_2$ (30 μM) was added to the culture medium 2 h before loading. In Supplementary Fig. 5, cells were co-pretreated with glutathione (1 mM) or PBN (10 mM) and L-OHP (100 μM) for 2 h. In Fig. 2a–c, mitoTEMPO (10 μM) was preloaded with Fura-2 loading. Note that all of the drugs used for pretreatments were washed out before Ca$^{2+}$ imaging experiments. Experiments were performed at room temperature unless otherwise stated. Cold stimulation was performed with an SC-20 dual in-line solution heater/cooler and a CL-100 temperature controller (Warner Instruments). The ratio of the fluorescence intensity obtained by the excitation/emission of 340 nm/510 nm (F$_{340}$) to the fluorescence intensity obtained by the excitation/emission of 380 nm/510 nm (F$_{380}$), namely F$_{340}$ F$_{380}^{-1}$, was calculated to quantify the intracellular Ca$^{2+}$ concentration. Cells with an F$_{340}$ F$_{380}^{-1}$ ratio >1.5 at baseline were excluded. Furthermore, in the experiments using cold stimulation, cells showing abnormal changes in the F$_{340}$ F$_{380}^{-1}$ ratio (that is, both F$_{340}$ and F$_{380}$ were increased or decreased simultaneously) following cold stimulation were also excluded. Statistical analysis of the change in the ratio, ΔRatio (F$_{340}$ F$_{380}^{-1}$), was performed as follows. In Figs 2c and 3b, the ΔRatio (F$_{340}$ F$_{380}^{-1}$) 2 min after cold stimulation was used. In Figs 2e,g, 3d, 5d,h and 6b,c, and Supplementary Figs 4 and 5, the mean of the maximum ΔRatio (F$_{340}$ F$_{380}^{-1}$) 5 min after the application of indicated TRPA1 agonist was used. AITC (100 μM) was used to validate the expression of TRPA1.

**Fluorometric $H_2O_2$ measurement.** Intracellular $H_2O_2$ levels were measured using PG-1, a fluorescent probe with high selectivity for $H_2O_2$ (refs 53,54). HEK293 cells on 35 mm dishes were loaded with PG-1 (5 μM) with or without mitoTEMPO (10 μM) for 40 min. Then the cells were placed on ice for 5 min. Then, the cells were collected using a cell scraper, lysed with 100 μl of DMSO and centrifuged at 7,000g for 5 min at 4 °C. The supernatants (80 μl) were loaded into a 96-well black plate, with 0.8 μl of PG-1 (1 mM) added. To remove background noise, cell-free blanks (80 μl of DMSO plus 0.8 μl of PG-1) were prepared and the average intensity of the blanks was subtracted from the samples. The PG-1 fluorescence was measured using a FLEX station (Molecular devices). Fluorescence measurements were performed at room temperature. The fluorescence intensity was obtained using an excitation/emission of 485 nm/525 nm.

**Western blotting.** Immunoblotting analyses were performed on whole-cell lysates. Cells grown in 35 mm dishes were treated with L-OHP (100 μM), DMO (30 μM) or $CoCl_2$ (100 μM) for 24 h at 37 °C and then lysed in a RIPA buffer. Aliquots of lysate were diluted with an equal volume of sample buffer containing (in mM) 100 Tris-HCl, 20% glycerol, 4% SDS, 12% 2-mercaptoethanol and 0.02% bromophenol blue, and loaded onto a 10% SDS-polyacrylamide gel. Proteins were blotted onto Immobilon-P PVDF transfer membranes (Millipore, Bedford, MA). The membranes were trimmed and exposed to a blocking solution and then incubated overnight at 4 °C with a monoclonal mouse antibody against HIF-1α (1:200, #NB100-105SS, Novus Biological, Littleton, CO) or a monoclonal mouse antibody against β-actin (1:10,000, #A-1978, Sigma). The following day, the membranes were briefly washed and then incubated with peroxidase-conjugated goat anti-mouse IgG (1:10,000, #115-035-003, GE Healthcare, Buckinghamshire, UK) for 1 h at room temperature. Specific bands were detected with Immobilon Western Chemiluminescent HRP Substrate (Millipore). Full blot image is shown in Supplementary Fig. 6.

**$H_2O_2$-evoked nocifensive behaviours.** Male mice were used for measurement of $H_2O_2$-evoked nocifensive behaviours. L-OHP, DMO, Pt(DACH)Cl$_2$ and DMOG were freshly dissolved in a 5% glucose solution. The HC030031 was dissolved in saline containing 3% DMSO, and $H_2O_2$ was diluted with saline. Mice received a single intraperitoneal (i.p.) administration of L-OHP (5 mg kg$^{-1}$), DMO (1.7 mg kg$^{-1}$), Pt(DACH)Cl$_2$ (4.8 mg kg$^{-1}$), DMOG (400 mg kg$^{-1}$) or vehicle 2 h before $H_2O_2$ (0.5%, 20 μl) was subcutaneously injected into the left hind paw (intraplantar). The doses of DMO and Pt(DACH)Cl$_2$ were calculated from the molecular weight of L-OHP. Vehicle or HC030031 (100 mg kg$^{-1}$) was i.p. administered 30 min before the injection of $H_2O_2$. The $H_2O_2$-evoked nocifensive behaviours were measured as durations of consecutive licking, flicking and biting behaviours for 5 min. Corresponding solvents were used for vehicle comparisons. All behaviours were tested by investigators blinded to the groups.

**Cold-plate test.** Male mice were used in cold-plate test. Cold sensitivity was assessed with a hot/cold-plate analgesiometer (Ugo Basile, Milan, Italy). Mice received a single i.p. administration of L-OHP (5 mg kg$^{-1}$), DMOG (400 mg kg$^{-1}$) or vehicle. After 1 h, the mice were allowed to acclimate to the testing apparatus for 1 h, after which they were individually placed in the centre of a cold-plate maintained at 5 °C in a transparent Plexiglas cylinder. The HC030031 (100 mg kg$^{-1}$), PBN (100 mg kg$^{-1}$) or vehicle was i.p. administered 30 min before tests. Escape behaviours were observed for 60 s and graded with a score of 0, 1 or 2 as follows: 0, no response; 1, moderate effort to avoid cold, such as lifting a hind paw or walking backwards; 2, vigorous effort to escape cold, such as jumping. The sum of the scores recorded within a 60 s period was calculated. All behaviours were tested by investigators blinded to the groups.

**Statistical analysis.** The data are presented as means ± s.e.m. from $n$ independent experiments. Statistical significances were calculated using GraphPad Prism 5 and 7 (GraphPad Software, La Jolla, CA). Differences between two groups were compared using unpaired Student's $t$-tests. Data from more than two groups were compared using one-way or two-way analyses of variance (ANOVA), followed by Tukey's or Sidak's multiple comparisons test or two-way ANOVA for repeated measures followed by the Bonferroni's *post hoc* test. In all cases, $P$ values < 5% ($P < 0.05$) were considered statistically significant.

**Data availability.** The data that support the findings of this study are available from the corresponding author on reasonable request.

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

## Acknowledgements

This work was supported in part by Grants-in-Aid for Scientific Research (KAKENHI) from the Japanese Society for the Promotion of Science (Grants-in-Aid for Scientific Research (B) to T.Na. (26293019) and S.K. (24390016), Challenging Exploratory Research to T.Na. (15K14961) and Scientific Research on Innovative Area 'Thermal Biology' to T.Na. (16H01386)), and by grants from the Salt Science Research Foundation (No. 14C4) and The Nakatomi Foundation. T.M. is a research fellow of Japan Society for the Promotion of Science.

## Author contributions

T.M., T.Na. and S.K. designed the project. T.M., S.N., M.Z., K.S., K.I. and N.T. performed the experiments. T.M., S.N., M.Z., K.S., H.S. and T.Na. analysed the data; T.Nu., N.T. and Y.M. provided materials and technical advices. T.M., T.Na. and S.K. wrote the manuscript. S.K. supervised the experiments and finalized the manuscript.

## Additional information

**Competing financial interests:** The authors declare no competing financial interests.

