## [Peer Review File · Nature Communications]

Reviewer #1 (Remarks to the Author)

In this manuscript the authors reveal a mechanism by which inhibition of hydroxylation and stimulation by reactive oxygen species combine to drastically potentiate the cold sensitivity of the chemosensory TRPA1 ion channel. This finding is significant as it resolves a long standing debate about the temperature-sensitivity of mammalian TRPA1 channels and their contribution to overall physiological temperature sensing. It also reveals an interesting allosteric mechanism of chemical and temperature gating, although this is not the focus of this work. I find the manuscript well written and data overall very convincing and have little to add, except for a few minor improvements:

- 1) The authors say that cold stimulation alone does not alter TRPA1 open probability, but that addition of H₂O₂ renders the channel more cold sensitive and the figure panel 1d certainly substantiates this claim. However, short representative recordings of single channel currents can be deceiving by not reflecting average open probabilities. A careful statistical analysis from long recordings of multiple patches is and statistical testing is therefore essential. In addition, it is important to perform this analysis also at a negative holding potential (e.g. -60 mV), which is the condition that is actually physiological relevant. The representative traces in Figure 1a and b suggest this should lead to an identical outcome, but careful analysis needs to be done.
- 2) I recommend adding a conceptual figure to help the reader understand the proposed mechanism and how it was probed experimentally.
- 3) The manuscript might benefit from a short paragraph discussing why exactly previous studies of the physiological role of TRPA1 as a cold sensor have come to drastically different conclusions.

Reviewer #2 (Remarks to the Author)

This manuscript by Miyata et al. provides a potential molecular explanation for the somewhat controversial nature of cold sensitivity of human TRPA1. They provide evidence that the cold sensitivity of this channel both in heterologous expression systems and in native mouse neurons is kept inherently low by hydroxylation of an N terminal domain proline residue, and that inhibition of this proline hydroxylation event allows cold-induced ROS generation to promote hTRPA1 activity in the presence of H₂O₂.

This is a compelling molecular explanation for a longstanding discrepancy in the field, and the authors have done a nice job of fleshing out their argument with cross-supportive pharmacological and genetic experiments. While the findings are generally convincing, however, there are several instances in which the conclusions are slightly overstated, and where the authors should exercise a bit more caution in their descriptions.

Specific comments:

- 1) Page 6: The authors claim that the demonstration of HIF 1 α induction by L-OHP or DMO supports the involvement of PHD in the ROS sensitization of hTRPA1 by L-OHP. This is a bit overstated. All it does is support the notion that PHD is inhibited, but does not further link PHD inhibition to hTRPA1 sensitization. That statement should be amended accordingly.
- 2) Figure 3e: Is the failure of L-OHP to sensitize hTRPA1 to H₂O₂ when wt PHD is overexpressed a function of incomplete enzyme inhibition and/or inadequate L-OHP concentration? This issue should be discussed when interpreting this experiment.
- 3) Page 6, last paragraph: The authors claim that the sensitization of hTRPA1 to cold by PHD represents a mechanism for peripheral neuropathy. However, the authors never study neuropathy in this manuscript. They look at a very acute behavioral response to L-OHP or oxalate exposure, but not neuropathy per se. This conclusion should be stated more carefully.

4) If the mouse data are to be interpreted in the context of the mechanisms studied in the rest of the manuscript, some corroborative experiments should be performed using mouse TRPA1.

Reviewer #3 (Remarks to the Author)

The study by Miyake et al. reports a novel regulation of TRPA1 by ROS that depends on hydroxylation state of a proline residue. If hydroxylated, ROS does not affect TRPA1 activity but dehydroxylated state allows ROS to activate TRPA1. The paper is clearly written and data seems robust. The results presented using appropriate statistical analysis support the existence of this type of regulation, and should be of interest to others in the field.

This work shows the regulation of TRPA1 by ROS in a PHD-dependent pathway and stops there. The work leaves the reader wondering about the physiological (not pharmacological) significance and the mechanism of ROS action. How does ROS activate TRPA1 via the proline residue? Under what circumstances is the proline hydroxylation altered to regulate TRPA1 activity via ROS? These may be difficult questions to address all at once, but some insight may help.

1. Page 5: "L-OHP-induced cold hypersensitivity depends on the ROS-mediated activation of TRPA1 sensitized by L-OHP"

Page 5: "ROS generated during L-OHP-pretreatment was dispensable for the induction of TRPA1 sensitization"

These two sentences seem contradictory. If the authors are correct, removal of H₂O₂ with antioxidants should remove cold hypersensitivity.

2. The increase in [H₂O₂] by cold seems rather small. According to the data, cold increases [ROS] by ~30%. Lets say that the [ROS] is ~10 nM at 37C and 13 nM at 16C. Could this small difference produce such a large effect on TRPA1 sensitivity? Is the basal level of H₂O₂ at 37C zero? If not, shouldn't the basal level of H₂O₂ also provide hypersensitivity or an increased sensitivity when proline is dehydroxylated?

3. Fig 1c/d shows that P-A mutant has a higher (almost double) basal TRPA1 activity at both 26C and 16C. This is not discussed at all.

4. Under normal conditions (without L-OHP), when would TRPA1 be not hydroxylated to provide cold hypersensitivity? Or is the observation simply a non-physiological one?

5. Does ROS cause hypersensitivity to other TRPA1 agonists?

6. Does extra [Ca] make a difference in the cold and ROS sensitivity of wild and mutant TRPA1?

7. Suppl Fig 7 should be moved to the main text, as this summarizes the data.

8. Fig 1d shows that H₂O₂ reduces TRPA1 mutant at 26C, but increases it at 16C. Also, it looks like the single channel amplitude is increased by H₂O₂ at 16C. These observations need to be clarified and explained.

9. Introduction: "However, despite the discovery of TRPA1 as a cold-activated channel², it is still debated whether TRPA1 is cold sensitive^{3,4} or not⁵⁻⁷. Recent studies report that rodent but not human TRPA1 is cold sensitive⁸, although purified human TRPA1 (hTRPA1) is intrinsically cold sensitive."

10. How is the finding related to the species difference? Do rodent TRPA1 also behave the same way with respect to ROS and cold (i.e., require proline dehydroxylation?) Behavioral experiments are done using rodents but electrophysiology is done using human TRPA1. Electrophysiology should also be done using rodents for proper comparison.

The comments from Reviewer #1 have been addressed as described below.

1) The authors say that cold stimulation alone does not alter TRPA1 open probability, but that addition of H₂O₂ renders the channel more cold sensitive and the figure panel 1d certainly substantiates this claim. However, short representative recordings of single channel currents can be deceiving by not reflecting average open probabilities. A careful statistical analysis from long recordings of multiple patches is and statistical testing is therefore essential. In addition, it is important to perform this analysis also at a negative holding potential (e.g. -60 mV), which is the condition that is actually physiological relevant. The representative traces in Figure 1a and b suggest this should lead to an identical outcome, but careful analysis needs to be done.

Responses; We agree the reviewer's advice that we need to perform statistical analysis of our single-channel experiments, thus we performed additional experiments. As shown in new Fig. 1f, g and Table 1, although hTRPA1-WT did not show any changes, hTRPA1-P394A showed an increase of open probability (NP_O) when they were exposed to the cold condition, but there is no significant difference between hTRPA1-WT and hTRPA1-P394A (new Table 1). However, additional H₂O₂ (0.1 μM) significantly augmented the NP_O of hTRPA1-P394A compared with hTRPA1-WT (new Table 1). Furthermore, consistent with the data of whole-cell patch clamp recordings (new Fig. 1b–d), only hTRPA1-P394A was sensitive to 0.1 μM H₂O₂ in the cold situation (16°C), but not at 26°C (new Fig. 1g). These results imply that relief from prolyl hydroxylation partially endows hTRPA1 with cold sensitivity, but it may not be enough to show a significant activation to trigger cold hypersensitivity *in vivo*. Instead, low concentration of ROS in the cold situation is important to drastically activate hTRPA1 lacking a

hydroxylation-susceptible proline residue. We added the results of statistical analysis to new Fig. 1g and Table 1, and corresponding description to the Result section from page 6 line 19 to page 7 line 8. To avoid misunderstandings, we changed the previous representative trace of hTRPA1-P394A to a reasonable one (new Fig. 1f).

Since we performed all of the excised patch recordings in the inside-out configuration, the holding potential (V_h) is the reciprocal number of the membrane potential (V_m). Thus, $V_h = +60$ mV is equal to $V_m = -60$ mV. However, considering

that the readers of *Nature Communications* are not always familiar with inside-out patch clamp recordings, the style using V_h may lead misunderstandings. Thus, we decided to use V_m instead of V_h in the current manuscript and changed some sentences of the Methods section in page 20 line 3 and 9 and the legend of new Fig. 1.

2) I recommend adding a conceptual figure to help the reader understand the proposed mechanism and how it was probed experimentally.

Responses; We agree the reviewer's suggestion and we added some illustrations in new Fig. 1 and Supplementary Figure 2 to help the reader understand our experimental conditions. Furthermore, we moved the conceptual figure to the main text (new Fig. 7).

3) The manuscript might benefit from a short paragraph discussing why exactly previous studies of the physiological role of TRPA1 as a cold sensor have come to drastically different conclusions.

Responses; Since our data represent the role of TRPA1 as a cold sensor in the L-OHP treated condition, we have not uncovered the physiological role of TRPA1 as a cold sensor yet. However, we suspect that the reasons why there are so many controversial conclusions of TRPA1 cold sensitivity may come from the view how ion channels sense environmental temperature. It is generally considered that thermal sensitivity of ion channels is intrinsic (Vriens, *et al.*, 2014, PMID: 25053448). In our paper, however, hTRPA1 is activated not by cold but by cold-produced ROS. In this point of view, it is not necessary that TRPA1 is intrinsically a cold-activated channel even if TRPA1 acts as a cold sensor in animals. We added the illustration of this point of view to the Discussion section from page 15 line 20 to page 16 line 1.

Furthermore, the procedure how to measure the channel activity may influence the conclusion. In our experiments, hTRPA1-P394A showed cold sensitivity in the

presence of H₂O₂ in the patch clamp recordings, but it also showed cold sensitivity even in the absence of H₂O₂ in the Ca²⁺ imaging experiments. Although we believe that these differences may be derived from the stability of [Ca²⁺]_i or the activity of intracellular organelle, further massive investigation will be required to solve this issue completely.

The comments from Reviewer #2 have been addressed as described below.

1) Page 6: The authors claim that the demonstration of HIF 1alpha induction by L-OHP or DMO supports the involvement of PHD in the ROS sensitization of hTRPA1 by L-OHP. This is a bit overstated. All it does is support the notion that PHD is inhibited, but does not further link PHD inhibition to hTRPA1 sensitization. That statement should be amended accordingly.

Responses; We agree with the reviewer's comment that our description is a bit overestimated, and we rewrote the sentence of the Result section from page 11 line 21 to page 12 line 2 to describe the results of Western blotting more precisely. Furthermore, we have rewritten the whole manuscript to adapt our previous manuscript (as Brief Communication in *Nature Neuroscience*) to the format of *Nature Communications*. Here, taken together with other data, we fully discuss the link between PHD inhibition and hTRPA1 sensitization.

2) Figure 3e: Is the failure of L-OHP to sensitize hTRPA1 to H₂O₂ when wt PHD is overexpressed a function of incomplete enzyme inhibition and/or inadequate L-OHP concentration? This issue should be discussed when interpreting this experiment.

Responses; We consider the failure of L-OHP to sensitize hTRPA1 to H₂O₂ when wt-PHD2 is overexpressed is caused by incomplete enzyme inhibition of L-OHP, as the reviewer mentioned. We have added the intention and interpretation of this experiment in the Result section in page 12 line 8–12. Thus, pretreatment of the cells with higher concentration of L-OHP may augment the hTRPA1 sensitivity to H₂O₂ even with the cells overexpressing wt-PHD2. However, we cannot examine this hypothesis because higher concentration of L-OHP ($\geq 300 \mu\text{M}$) immediately activates hTRPA1 via immediate ROS generation as previously described (Nassini, *et al.*, 2011, PMID: 21481532).

3) Page 6, last paragraph: *The authors claim that the sensitization of hTRPA1 to cold by PHD represents a mechanism for peripheral neuropathy. However, the authors never study neuropathy in this manuscript. They look at a very acute behavioral response to L-OHP or oxalate exposure, but not neuropathy per se. This conclusion should be stated more carefully.*

Responses; We agree the reviewer's comment that our data did not support the L-OHP-induced neuropathy. The L-OHP-induced neuropathy is known to be categorized into 2 types; acute but reversible peripheral neuropathy, in which the symptoms include cold hypersensitivity, and chronic and cumulative form (McWhinney, *et al.*, 2009, PMID:19139108). Although the former acute form is specific to L-OHP, the latter chronic form is more general and may be usually called as chemotherapeutic-induced peripheral neuropathy, because it is also observed in the patients treated with other chemotherapeutic agents such as paclitaxel and cisplatin. However, to avoid the misinterpretation of the readers, we used the term "cold hypersensitivity" instead of "neuropathy" in the current manuscript. We added the explanation of L-OHP-induced acute cold hypersensitivity to the Introduction section in page 5 line 6–8 and the Discussion section in page 14 line 19–20.

4) *If the mouse data are to be interpreted in the context of the mechanisms studied in the rest of the manuscript, some corroborative experiments should be performed using mouse TRPA1.*

Responses; We agree the reviewer's idea that some corroborative experiments using mTRPA1 is required, and we performed inside-out patch clamp recordings and Ca^{2+} imaging experiments using mTRPA1-expressing HEK293 cells. As shown in new Fig. 3, mTRPA1 behaved similarly to hTRPA1. Although mTRPA1-expressing cells responded to cold stimulation significantly larger than hTRPA1-expressing cells ($P < 0.001$, compared with the data of non-treated hTRPA1-WT expressing cells in the new Fig. 2c, unpaired *t*-test), DMOG pretreatment further augmented the cold-induced $[\text{Ca}^{2+}]_i$ increase, which was significantly suppressed in the presence of a ROS scavenger PBN (new Fig. 3a, b). We chose PBN in this experiment because we used it in the *in vivo* experiments. Consistent with this result, our inside-out patch clamp recordings revealed that DMOG-pretreated mTRPA1 was significantly activated by 0.1 μM H_2O_2

compared with non-treated mTRPA1 in the cold situation (new Fig. 3c). Furthermore, consistent with the data of hTRPA1 (new Fig. 2d, e), mTRPA1-expressing cells pretreated with DMOG showed a significant increase of sensitivity to H₂O₂ (new Fig. 3d). These results indicate that PHD inhibition induced an enhancement of sensitivity to H₂O₂ in both hTRPA1 and mTRPA1. We added the new results in Fig. 3 and corresponding description to the Result section in page 9 line 1–12 to the Discussion section in page 14 line 8–14.

The comments from Reviewer #3 have been addressed as described below.

1) Page 5: "L-OHP-induced cold hypersensitivity depends on the ROS-mediated activation of TRPA1 sensitized by L-OHP"

Page 5: "ROS generated during L-OHP-pretreatment was dispensable for the induction of TRPA1 sensitization"

These two sentences seem contradictory. If the authors are correct, removal of H₂O₂ with antioxidants should remove cold hypersensitivity.

Responses; We agree the reviewer's question that removal of H₂O₂ with antioxidants attenuates cold hypersensitivity, and in fact, pre-administration of a ROS scavenger PBN significantly suppressed DMOG- and L-OHP-induced cold hypersensitivity in mice (new Fig. 4a, b). In new Supplementary Figure 5a, we co-pretreated the cells with ROS scavengers and L-OHP for 2 h. Then, all of the drugs were washed out and we performed the Ca²⁺ imaging experiments. In this experiment, ROS scavengers had no effect on the induction of L-OHP-induced TRPA1 sensitization to H₂O₂ (new Supplementary Figure 5a). These results indicate that ROS presumably produced during the L-OHP pretreatment is not necessary for the induction of TRPA1 sensitization, but ROS produced during cold stimulation is important to elicit cold hypersensitivity. To avoid misleading caused by our poor description, we added note in the legend of Supplementary Figure 5a.

2) The increase in [H₂O₂] by cold seems rather small. According to the data, cold increases [ROS] by ~30%. Lets say that the [ROS] is ~10 nM at 37°C and 13 nM at 16°C. Could this small difference produce such a large effect on TRPA1 sensitivity? Is the basal

level of H₂O₂ at 37°C zero? If not, shouldn't the basal level of H₂O₂ also provide hypersensitivity or an increased sensitivity when proline is dehydroxylated?

Responses; It is very difficult to measure the intracellular H₂O₂ concentration ([H₂O₂]_i) to determine whether the increase can reach sufficient to stimulate the sensitized hTRPA1 in this method, because PG-1 reacts to H₂O₂ irreversibly, although it has high specificity to H₂O₂. To observe the Δ [H₂O₂]_i induced by cold stimulation, we compared the samples with the samples collected before cold stimulation. Although the samples incubated at 37°C showed an increase of the fluorescence of PG-1 (F_{PG-1}), but it did not reach significance (new Supplementary Figure 3). Only the samples received cold stimulation showed a significant increase of F_{PG-1} (new Supplementary Figure 3). Furthermore, we also observed that these increase of F_{PG-1} were decreased when cells were treated with mitoTEMPO (new Supplementary Figure 3). These results indicate that cold stimulation induces presumably mitochondria-derived H₂O₂ generation, but we could not fully exclude the possibility that basal level of H₂O₂ also contributes to the observed cold hypersensitivity of TRPA1 (not the hypersensitivity to ROS), as the reviewer mentioned. We added this result to Supplementary Figure 3, corresponding illustration to the Result section in page 7 line 19–21, and the possibility to the Discussion section in page 14 line 14–16.

However, we could not measure the precise Δ [H₂O₂]_i increase in the above experiments. To respond the reviewer's comment partly, we further performed whole-cell patch clamp recordings in the presence of lower concentration of H₂O₂ to estimate the lower limit of Δ [H₂O₂]_i increase required for hTRPA1 activation in the cold situation. We found that both hTRPA1-WT and hTRPA1-P394A did not show the cold-induced activation in the presence of 0.03 μ M H₂O₂ (hTRPA1-WT; 1.844 ± 3.889 pA/pF, hTRPA1-P394A; 3.092 ± 4.158 pA/pF, $p = 0.829$, unpaired t -test, $n = 8-9$ cells, data are Δ current at +80 mV and expressed as means \pm s.e.m.). Although these data are not shown in the manuscript, these results imply that cold-induced Δ [H₂O₂]_i increase is quite tiny, but it will be more than 0.03 μ M, considering that hTRPA1-P394A or DMOG-pretreated hTRPA1-WT is activated by cold-induced ROS in the Ca²⁺ imaging experiments (new Fig. 2a-c).

3) Fig 1c/d shows that P-A mutant has a higher (almost double) basal TRPA1 activity at both 26°C and 16°C. This is not discussed at all.

Responses; In the new manuscript, we performed statistical analysis of our single-channel experiments and described this result in new Table 1. According to the new results, the basal activity of hTRPA1-P394A is larger than that of hTRPA1-WT both at 26°C and 16°C, but it did not reach statistical significance (new Table 1). This may be derived from the difference of basal activity between hTRPA1-WT and hTRPA1-P394A, as previously described (Takahashi, et al., 2011, PMID: 21873995). We added these results in Table 1 and corresponding explanation to the Result section from page 6 line 19 to page 7 line 3. To avoid misinterpretations, we changed the representative trace of hTRPA1-P394A in new Fig. 1f.

4) Under normal conditions (without L-OHP), when would TRPA1 be not hydroxylated to provide cold hypersensitivity? Or is the observation simply a non-physiological one?

Responses; Although we observed only drug-induced cold hypersensitivity in this study, considering that prolyl hydroxylation is regulated by PHDs that are the well-known O₂ sensing molecules, we guess that the mechanisms we uncovered in this study underlie other cold hypersensitivity triggered by peripheral ischemia. In fact, it is reported that patients with peripheral ischemic diseases such as Buerger's disease (Dargon, *et al.*, 2012, PMID: 22284771) and Raynaud syndrome (Valdovinos, *et al.*, 2014, PMID: 25770637) are known to complain cold hypersensitivity. We mentioned our perspective in the Discussion section in page 15 line 15–19.

5) Does ROS cause hypersensitivity to other TRPA1 agonists?

Responses; Although our poor description leads the reviewer's misunderstanding, we believe that PHD-inhibition, but not ROS, causes sensitization of TRPA1. Thus, we explored whether the PHD-inhibition induced hypersensitivity to other TRPA1 agonists. As shown in new Supplementary Figure 4, we tested three well-known hTRPA1 agonists (AITC, 2-APB, and menthol). We found that hTRPA1-P394A showed significantly high sensitivity to only AITC, but not 2-APB and menthol. It is known that AITC activates TRPA1 through modification of N-terminal cysteine residues of TRPA1, while 2-APB and menthol activate TRPA1 in other mechanisms (Hinman, et al., 2006, PMID: 17164327; Xiao, et al., 2007, PMID: 18815250). Considering that H₂O₂ also activates TRPA1 through cysteine modification (Takahashi, et al., 2008, PMID: 18769139), PHD-inhibition may induce hypersensitivity to only cysteine-dependent

TRPA1 agonists. We added the new results to new Supplementary Figure 4 and corresponding description to the Result section in page 8 line 12–21.

6) Does extra [Ca] make a difference in the cold and ROS sensitivity of wild and mutant TRPA1?

Responses; To answer the reviewer's question, we performed whole-cell patch clamp recordings using Ca^{2+} -free extracellular solution. As shown in new Supplementary Figure 2, hTRPA1-P394A did not show the cold-induced currents in this condition ($n = 6$ cells). Since permeating Ca^{2+} ions are reported to potentiate TRPA1 in whole-cell patch clamp recordings even in the presence of 5 mM BAPTA in the pipette (Wang, et al., 2008, PMID: 18775987), the basal Ca^{2+} flux through TRPA1 may be also necessary to potentiate TRPA1 enough to sense cold-induced ROS generation. We added the new results to new Supplementary Figure 2 and corresponding description to the Result section in page 6 line 13–17.

7) Suppl Fig 7 should be moved to the main text, as this summarizes the data.

Responses; We agree the reviewer's suggestion and moved the conceptual figure to the main text as new Fig. 7.

8) Fig 1d shows that H_2O_2 reduces TRPA1 mutant at 26°C, but increases it at 16°C. Also, it looks like the single channel amplitude is increased by H_2O_2 at 16°C. These observations need to be clarified and explained.

Responses; We performed additional single-channel experiments. The statistical analysis indicate that the NP_0 of hTRPA1-P394A showed a bit increase by application of H_2O_2 at 26°C, and showed a huge increase by application of H_2O_2 at 16°C (new Supplementary Table 1). To avoid the reader's misunderstandings, we changed the representative trace of hTRPA1-P394A (new Fig. 1f). We further investigated the single channel conductance and found that the single channel conductance of hTRPA1-P394A was significantly increased in the presence of H_2O_2 at 16°C (new Table 1). Since it is reported that hTRPA1 is dilated when it is activated by agonists such as AITC (Bobkov, et al., 2011, PMID: 21195050), these results may indicate that both cold and H_2O_2 are necessary for hTRPA1-P394A to be activated. We added these results in Table 1 and corresponding illustration to the Result section in page 7 line 3–8.

9. Introduction: "However, despite the discovery of TRPA1 as a cold-activated channel², it is still debated whether TRPA1 is cold sensitive^{3,4} or not⁵⁻⁷. Recent studies report that rodent but not human TRPA1 is cold sensitive⁸, although purified human TRPA1 (hTRPA1) is intrinsically cold sensitive." How is the finding related to the species difference? Do rodent TRPA1 also behave the same way with respect to ROS and cold (i.e., require proline dehydroxylation?) Behavioral experiments are done using rodents but electrophysiology is done using human TRPA1. Electrophysiology should also be done using rodents for proper comparison.

Responses; To answer the reviewer's question, we performed inside-out patch clamp recordings using mTRPA1. As shown in new Fig. 3, Ca²⁺ imaging experiments revealed that DMOG pretreatment augmented the cold-induced [Ca²⁺]_i increase, which was significantly suppressed in the presence of a ROS scavenger PBN (new Fig. 3a, b). We used PBN in this experiment because we used it in the *in vivo* experiments. Similarly, inside-out patch clamp recordings revealed that DMOG-pretreated mTRPA1 showed a significant increase of NP_O triggered by 0.1 μM H₂O₂ compared with non-treated mTRPA1 in the cold situation (new Fig. 3c). Furthermore, mTRPA1-expressing cells pretreated with DMOG showed a significant increase of sensitivity to H₂O₂ (new Fig. 3d). These results indicate that the sensitivity of both hTRPA1 and mTRPA1 to H₂O₂ is enhanced by PHD inhibition. Since the Ca²⁺ response of mTRPA1-expressing cells induced by cold stimulation was significantly larger than that of hTRPA1-expressing cells ($P < 0.001$, the data of non-treated mTRPA1 expressing cells in the new Fig. 3b was compared with that of non-treated hTRPA1-WT expressing cells in the new Fig. 2c, unpaired *t*-test) but H₂O₂ (10 μM)-induced Ca²⁺ response was similar in both species (for example, compared the data of non-treated mTRPA1 expressing cells in the new Fig. 3d with that of non-treated hTRPA1 expressing cells in the new Fig. 2e), we considered that the molecular details of intrinsic cold sensitivity of mTRPA1 is different from the mechanisms we revealed in this study. We added the new results in Fig. 3 and corresponding description to the Result section in page 9 line 1–12 and to the Discussion section in page 14 line 8–14.

Reviewer #1 (Remarks to the Author)

All my comments have been addressed. This is now a very nice manuscript.

Reviewer #2 (Remarks to the Author)

The authors have very thoroughly addressed all of this reviewer's criticisms. The manuscript is much improved.

Reviewer #3 (Remarks to the Author)

Authors have done a thorough job of responding to reviewers' comments. The findings are original, methods and quality of data are good and conclusion is robust. I just have following minor points

"The hTRPA1-P394A was not activated by a combined stimulation of cold and H₂O₂ (Supplementary Fig. 2), suggesting that an enhanced effect of extracellular Ca²⁺ ions was also involved."---This important point is skimmed over and not followed up. Influx of Ca²⁺ is clearly necessary for TRPA1 activation by ROS--what is Ca²⁺ doing? Some discussion is warranted because this is an important point. Is Ca²⁺ necessary for ROS production?

"In the absence of H₂O₂, cold stimulation (from 26{degree sign}C to 16{degree sign}C) failed to increase the NPO of hTRPA1-WT, but significantly increased the NPO of hTRPA1-P394A, although it did not reach statistical significance when compared with hTRPA1-WT." --if significant, how could it not reach statistical significance? Does not make sense.

"Single channel conductance of hTRPA1-P394A was also significantly larger in the presence of 0.1 μM H₂O₂ only at 16{degree sign}C"--larger than what??

MitoTEMPO was preloaded with PG-1; what does this mean?

"we next examined the altered response of hTRPA1 to ROS is solely caused by inhibiting"--grammar problem.

The cold-induced production of ROS could be much higher in native cells than in HEK cells. It would be interesting to test this, as ROS is critical for TRPA1 cold response.

One thing that still bothers me is the finding that cold increases [ROS]. How would this occur? Many biochemical reactions are slowed at cold temp.

The comments from Reviewer #3 have been addressed as described below:

"The hTRPA1-P394A was not activated by a combined stimulation of cold and H₂O₂ (Supplementary Fig. 2), suggesting that an enhanced effect of extracellular Ca²⁺ ions was also involved."---This important point is skimmed over and not followed up. Influx of Ca²⁺ is clearly necessary for TRPA1 activation by ROS--what is Ca²⁺ doing? Some discussion is warranted because this is an important point. Is Ca²⁺ necessary for ROS production?

Response; Although we used a fast Ca²⁺ chelator BAPTA (5 mM) in the whole-cell patch clamp recordings to buffer intracellular Ca²⁺ ion and minimize the effect of Ca²⁺ influx on intracellular Ca²⁺ concentration, we could not fully exclude the effects of Ca²⁺ influx on ROS production because mitochondrial matrix Ca²⁺ overload is known to cause ROS generation (Brookes, *et al.*, PMID: 15355853). Considering that hTRPA1-P394A showed an increase of NP_O when stimulated by cold and H₂O₂ in the inside-out patch clamp recordings (Table 1), which is also under an extracellular Ca²⁺ free condition, the reason of the lack of the cold- and H₂O₂-induced whole-cell currents in the extracellular Ca²⁺ free condition may be derived from the inefficiency of ROS generation from mitochondria. However, we also observed that cold stimulation without H₂O₂ failed to evoke whole-cell current in hTRPA1-P394A-expressing cells even under 2 mM extracellular Ca²⁺ condition, although the same stimulation increased the NP_O in an inside-out patch clamp under extracellular Ca²⁺ free condition (Table 1). This difference may come from the difference of the sensitivity between these recordings, implying that extracellular Ca²⁺ ion is necessary to amplify the hTRPA1-P394A currents enough to be detected by whole-cell patch clamp recordings. Furthermore, we have not yet compared the Ca²⁺ sensitivity between hTRPA1-WT and hTRPA1-P394A. Thus, we need further precise investigation to clarify the role of Ca²⁺ in this situation in the future. We mentioned this point in the Discussion section page 14, line 19–22 and added one new reference.

"In the absence of H₂O₂, cold stimulation (from 26°C to 16°C) failed to increase the NP_O of hTRPA1-WT, but significantly increased the NP_O of hTRPA1-P394A, although it did not

reach statistical significance when compared with hTRPA1-WT." --if significant, how could it not reach statistical significance? Does not make sense.

Response; We feel sorry for our poor description. The NP_O of hTRPA1-P394A at 16°C was larger than the NP_O of hTRPA1-P394A at 26°C (basal), but it did not reach statistical significance when compared with that of hTRPA1-WT at 16°C. We revised the sentence in the Result section page 7, line 1–3.

"Single channel conductance of hTRPA1-P394A was also significantly larger in the presence of 0.1 μM H₂O₂ only at 16°C"--larger than what??"

Response; We feel sorry for forgetting to state the comparison. Single channel conductance of hTRPA1-P394A in the presence of H₂O₂ at 16°C was significantly larger than its basal level (26°C, no H₂O₂). We revised the sentence in the Result section page 7, line 6–7.

MitoTEMPO was preloaded with PG-1; what does this mean?

Response; Since mitoTEMPO is a mitochondria-targeted antioxidant, it requires loading time to accumulate in mitochondria. We loaded mitoTEMPO with PG-1 for 40 min to clarify whether cold-evoked ROS is mitochondria-derived or not, but we could not exclude the possibility that mitoTEMPO preloading has effect on the basal [H₂O₂]_i level, as previously the reviewer mentioned. We added our interpretation in the Result section from page 7, line 22 to page 8, line 2.

"we next examined the altered response of hTRPA1 to ROS is solely caused by inhibiting"--grammar problem.

Response; We feel sorry for our poor English skill and corrected the mistake in the Result section page 8, line 9–10.

The cold-induced production of ROS could be much higher in native cells than in HEK cells. It would be interesting to test this, as ROS is critical for TRPA1 cold response.

Response; To answer the reviewer's interest, we show our preliminary data for the production of ROS using mice *in vivo*, because of the difficulty of preparing enough amount of native cells *in vitro* (80-90% confluence in a 35 mm dish, four dishes are at least required to get a data of one condition). We intraplantarly (i.pl.) administrated PG-1 to the both side of mouse paws, and 9 h later, i.pl. H₂O₂ injection or cold stimulation was performed to the one side of paws. After 10 or 20 min, we sacrificed the mice and observed

the PG-1 fluorescence (we waited 20 min in the cold stimulated experiment to acquire enough amount of the irreversible PG-1 oxidized product). The results are attached the bottom of this letter. The sample received i.pl. H₂O₂ group showed a punctual pattern of PG-1 fluorescence in the ipsilateral side, but not in the contralateral side. However, we could not get any PG-1 fluorescence from the sample received cold stimulated group. These results indicate that i.pl. PG-1 can detect H₂O₂, but it seems to require relatively large amount of H₂O₂ and we cannot detect the cold-evoked ROS generation that seems very small compared to the H₂O₂ injection. Considering the previous reports (as discussed below), we believe that cold stimulation can induce mitochondrial ROS generation, but we need further technical evolution to detect such a localized ROS generation *in vivo*.

One thing that still bothers me is the finding that cold increases [ROS]. How would this occur? Many biochemical reactions are slowed at cold temp.

Response; The molecular mechanisms how cold stimulation triggers ROS generation from mitochondria still remains to be unsolved. In addition to the report we cited in the present manuscript (Bailey *et al.*, PMID: 15764673), recently, Chouchani *et al.* reported that cold (4°C) exposure acutely activated brown adipose tissue (BAT) thermogenesis in mice, which was associated with a substantial increase in mitochondrial ROS and substantial oxidation, and depletion of the BAT glutathione pool (PMID: 27027295). Furthermore, this report also showed that scavenging mitochondrial ROS resulted in hypothermia upon cold exposure in a uncoupling protein 1 (UCP1) dependent manner, indicating that mitochondrial ROS and UCP1 have a pivotal role in thermo generation in the cold situation. Thus, cold-evoked mitochondrial ROS generation seems to have a physiologically important role in thermo generation to resist cold exposure. Although we need further multiple approaches, it is worth answering the question how cold stimulation trigger mitochondrial ROS generation. We presented this discussion in the Discussion section page 16, line 8–11 and added one new reference.